# CEM-RL: Combining evolutionary and gradient-based methods for policy search

**Aloïs Pourchot**[1,2]**, Olivier Sigaud**[2]

(1) Gleamer
96bis Boulevard Raspail, 75006 Paris, France
`alois.pourchot@gleamer.ai`

(2) Sorbonne Université, CNRS UMR 7222
Institut des Systèmes Intelligents et de Robotique, F-75005 Paris, France
`olivier.sigaud@upmc.fr`   +33 (0) 1 44 27 88 53

## Abstract

Deep neuroevolution and deep reinforcement learning (deep RL) algorithms are two popular approaches to policy search. The former is widely applicable and rather stable, but suffers from low sample efficiency. By contrast, the latter is more sample efficient, but the most sample efficient variants are also rather unstable and highly sensitive to hyper-parameter setting. So far, these families of methods have mostly been compared as competing tools. However, an emerging approach consists in combining them so as to get the best of both worlds. Two previously existing combinations use either an ad hoc evolutionary algorithm or a goal exploration process together with the Deep Deterministic Policy Gradient (DDPG) algorithm, a sample efficient off-policy deep RL algorithm. In this paper, we propose a different combination scheme using the simple cross-entropy method (CEM) and Twin Delayed Deep Deterministic policy gradient (TD3), another off-policy deep RL algorithm which improves over DDPG. We evaluate the resulting method, CEM-RL, on a set of benchmarks classically used in deep RL. We show that CEM-RL benefits from several advantages over its competitors and offers a satisfactory trade-off between performance and sample efficiency.

## 1 Introduction

Policy search is the problem of finding a policy or controller maximizing some unknown utility function. Recently, research on policy search methods has witnessed a surge of interest due to the combination with deep neural networks, making it possible to find good enough continuous action policies in large domains. From one side, this combination gave rise to the emergence of efficient deep reinforcement learning (deep RL) techniques (Lillicrap et al., 2015; Schulman et al., 2015; 2017). From the other side, evolutionary methods, and particularly deep neuroevolution methods applying evolution strategies (ESs) to the parameters of a deep network have emerged as a competitive alternative to deep RL due to their higher parallelization capability (Salimans & Kingma, 2016; Conti et al., 2017; Petroski Such et al., 2017).

Both families of techniques have clear distinguishing properties. Evolutionary methods are significantly less sample efficient than deep RL methods because they learn from complete episodes, whereas deep RL methods use elementary steps of the system as samples, and thus exploit more information (Sigaud & Stulp, 2018). In particular, off-policy deep RL algorithms can use a replay buffer to exploit the same samples as many times as useful, greatly improving sample efficiency. Actually, the sample efficiency of ESs can be improved using the "importance mixing" mechanism, but a recent study has shown that the capacity of importance mixing to improve sample efficiency by a factor of ten is still not enough to compete with off-policy deep RL (Pourchot et al., 2018). From the other side, sample efficient off-policy deep RL methods such as the DDPG algorithm (Lillicrap et al., 2015) are known to be unstable and highly sensitive to hyper-parameter setting. Rather than opposing both families as competing solutions to the policy search problem, a richer perspective

consists in combining them so as to get the best of both worlds. As covered in Section 2, there are very few attempts in this direction so far.

After presenting some background in Section 3, we propose in Section 4 a new combination method that combines the cross-entropy method (CEM) with DDPG or TD3, an off-policy deep RL algorithm which improves over DDPG. In Section 5, we investigate experimentally the properties of this CEM-RL method, showing its advantages both over the components taken separately and over a competing approach. Beyond the results of CEM-RL, the conclusion of this work is that there is still a lot of unexplored potential in new combinations of evolutionary and deep RL methods.

## 2   RELATED WORK

Policy search is an extremely active research domain. The realization that evolutionary methods are an alternative to continuous action reinforcement learning and that both families share some similarity is not new (Stulp & Sigaud, 2012a;b; 2013) but so far most works have focused on comparing them (Salimans et al., 2017; Petroski Such et al., 2017; Conti et al., 2017). Under this perspective, it was shown in (Duan et al., 2016) that, despite its simplicity with respect to most deep RL methods, the Cross-Entropy Method (CEM) was a strong baseline in policy search problems. Here, we focus on works which combine both families of methods.

Synergies between evolution and reinforcement learning have already been investigated in the context of the so-called *Baldwin effect* (Simpson, 1953). This literature is somewhat related to research on meta-learning, where one seeks to evolve an initial policy from which a self-learned reinforcement learning algorithm will perform efficient improvement (Wang et al., 2016; Houthooft et al., 2018; Gupta et al., 2018). The key difference with respect to the methods investigated here is that in this literature, the outcome of the RL process is not incorporated back into the genome of the agent, whereas here evolution and reinforcement learning update the same parameters in iterative sequences.

Closer to ours, the work of Colas et al. (2018) sequentially applies a *goal exploration process* (GEP) to fill a replay buffer with purely exploratory trajectories and then applies DDPG to the resulting data. The GEP shares many similarities with evolutionary methods, though it focuses on diversity rather than on performance of the learned policies. The authors demonstrate on the Continuous Mountain Car and HALF-CHEETAH-V2 benchmarks that their combination, GEP-PG, is more sample-efficient than DDPG, leads to better final solutions and induces less variance during learning. However, due to the sequential nature of the combination, the GEP part does not benefit from the efficient gradient steps of the deep RL part.

Another approach related to ours is the work of Maheswaranathan et al. (2018), where the authors introduce optimization problems with a surrogate gradient, i.e. a direction which is correlated with the real gradient. They show that by modifying the covariance matrix of an ES to incorporate the informations contained in the surrogate, a hybrid algorithm can be constructed. They provide a thorough theoretical investigation of their procedure, which they experimentally show capable of outperforming both a standard gradient descent method and a pure ES on several simple benchmarks. They argue that this method could be useful in RL, since surrogate gradients appear in Q-learning and actor-critic methods. However, a practical demonstration of those claims remains to be performed. Their approach resembles ours in that they use a gradient method to enhance an ES. But a notable difference is that they use the gradient information to directly change the distribution from which samples are drawn, whereas we use gradient information on the samples themselves, impacting the distribution only indirectly.

The work which is the closest to ours is Khadka & Tumer (2018b). The authors introduce an algorithm called ERL (for Evolutionary Reinforcement Learning), which is presented as an efficient combination of a deep RL algorithm, DDPG, and a population-based evolutionary algorithm. It takes the form of a population of actors, which are constantly mutated and selected in tournaments based on their fitness. In parallel, a single DDPG agent is trained from the samples generated by the evolutionary population. This single agent is then periodically inserted into the population. When the gradient-based policy improvement mechanism of DDPG is efficient, this individual outperforms its evolutionary siblings, it gets selected into the next generation and draws the whole population towards higher performance. Through their experiments, Khadka & Tumer demonstrate that this setup

benefits from an efficient transfer of information between the RL algorithm and the evolutionary algorithm, and vice versa.

However, their combination scheme does not make profit of the search efficiency of ESs. This is unfortunate because ESs are generally efficient evolutionary methods, and importance mixing can only be applied in their context to bring further sample efficiency improvement.

By contrast with the works outlined above, the method presented here combines CEM and TD3 in such a way that our algorithm benefits from the gradient-based policy improvement mechanism of TD3, from the better stability of ESs, and may even benefit from the better sample efficiency brought by importance mixing, as described in Appendix B.

## 3 BACKGROUND

In this section, we provide a quick overview of the evolutionary and deep RL methods used throughout the paper.

### 3.1 EVOLUTIONARY ALGORITHMS, EVOLUTION STRATEGIES AND EDAS

Evolutionary algorithms manage a limited population of individuals, and generate new individuals randomly in the vicinity of the previous *elite* individuals. There are many variants of such algorithms, some using tournament selection as in Khadka & Tumer (2018b), niche-based selection or more simply taking a fraction of elite individuals, see Back (1996) for a broader view. Evolution strategies can be seen as specific evolutionary algorithms where only one individual is retained from one generation to the next, this individual being the mean of the distribution from which new individuals are drawn. More specifically, an optimum individual is computed from the previous samples and the next samples are obtained by adding Gaussian noise to the current optimum. Finally, among ESs, Estimation of Distribution Algorithms (EDAs) are a specific family where the population is represented as a distribution using a covariance matrix $\Sigma$ (Larrañaga & Lozano, 2001). This covariance matrix defines a multivariate Gaussian function and samples at the next iteration are drawn according to $\Sigma$. Along iterations, the ellipsoid defined by $\Sigma$ is progressively adjusted to the top part of the hill corresponding to the local optimum $\theta^*$. Various instances of EDAs, such as the Cross-Entropy Method (CEM), Covariance Matrix Adaptation Evolution Strategy (CMA-ES) and $\text{PI}^2$-CMA, are covered in Stulp & Sigaud (2012a;b; 2013). Here we focus on the first two.

### 3.2 THE CROSS-ENTROPY METHOD AND CMA-ES

The Cross-Entropy Method (CEM) is a simple EDA where the number of elite individuals is fixed to a certain value $K_e$ (usually set to half the population). After all individuals of a population are evaluated, the $K_e$ fittest individuals are used to compute the new mean and variance of the population, from which the next generation is sampled after adding some extra variance $\epsilon$ to prevent premature convergence.

In more details, each individual $x_i$ is sampled by adding Gaussian noise around the mean of the distribution $\mu$, according to the current covariance matrix $\Sigma$, i.e. $x_i \sim \mathcal{N}(\mu, \Sigma)$. The problem-dependent fitness of these new individuals $(f_i)_{i=1,\dots,\lambda}$ is computed, and the top-performing $K_e$ individuals, $(z_i)_{i=1,\dots,K_e}$ are used to update the parameters of the distribution as follows:

$$\mu_{new} = \sum_{i=1}^{K_e} \lambda_i z_i \tag{1}$$

$$\Sigma_{new} = \sum_{i=1}^{K_e} \lambda_i (z_i - \mu_{old})(z_i - \mu_{old})^T + \epsilon \mathcal{I}, \tag{2}$$

where $(\lambda_i)_{i=1,\dots,K_e}$ are weights given to the individuals, commonly chosen with $\lambda_i = \frac{1}{K_e}$ or $\lambda_i = \frac{log(1+K_e)/i}{\sum_{i=1}^{K_e} \log(1+K_e)/i}$ (Hansen, 2016). In the former, each individual is given the same importance, whereas the latter gives more importance to better individuals.

Similarly to CEM, Covariance Matrix Adaptation Evolution Strategy (CMA-ES) is an EDA where the number of elite individuals is fixed to a certain value $K_e$. The mean and covariance of the new generation are constructed from those individuals. However this construction is more elaborate than in CEM. The top $K_e$ individuals are ranked according to their performance, and are assigned weights based on this ranking. Those weights in turn impact the construction of the new mean and covariance. Quantities called "Evolutionary paths" are also used to accumulate the search directions of successive generations. In fact, the updates in CMA-ES are shown to approximate the natural gradient, without explicitly modeling the Fisher information matrix (Arnold et al., 2011).

A minor difference between CEM and CMA-ES can be found in the update of the covariance matrix. In its standard formulation, CEM uses the new estimate of the mean $\mu$ to compute the new $\Sigma$, whereas CMA-ES uses the current $\mu$ (the one that was used to sample the current generation) as is the case in (2). We used the latter as Hansen (2016) shows it to be more efficient. The algorithm we are using can thus be described either as CEM using the current $\mu$ for the estimation of the new $\Sigma$, or as CMA-ES without evolutionary paths. The difference being minor, we still call the resulting algorithm CEM. Besides, we add some noise in the form of $\epsilon \mathcal{I}$ to the usual covariance update to prevent premature convergence. We choose to have an exponentially decaying $\epsilon$, by setting an initial and a final standard deviation, respectively $\sigma_{init}$ and $\sigma_{end}$, initializing $\epsilon$ to $\sigma_{init}$ and updating $\epsilon$ at each iteration with $\epsilon = \tau_{cem}\epsilon + (1 - \tau_{cem})\sigma_{end}$.

Note that, in practice $\Sigma$ can be too large for computing the updates and sampling new individuals. Indeed, if $n$ denotes the number of actor parameters, simply sampling from $\Sigma$ scales at least in $\mathcal{O}(n^{2.3})$, which becomes quickly intractable. Instead, we constrain $\Sigma$ to be diagonal. This means that in our computations, we replace the update in (2) by

$$\Sigma_{new} = \sum_{i=1}^{K_e} \lambda_i (z_i - \mu_{old})^2 + \epsilon \mathcal{I}, \tag{3}$$

where the square of the vectors denote the vectors of the square of the coordinates.

## 3.3 DDPG AND TD3

The Deep Deterministic Policy Gradient (DDPG) (Lillicrap et al., 2015) and Twin Delayed Deep Deterministic policy gradient (TD3) (Fujimoto et al., 2018) algorithms are two off-policy, actor-critic and sample efficient deep RL algorithms. The DDPG algorithm suffers from instabilities partly due to an overestimation bias in critic updates, and is known to be difficult to tune given its sensitivity to hyper-parameter settings. The availability of properly tuned code baselines incorporating several advanced mechanisms improves on the latter issue (Dhariwal et al., 2017). The TD3 algorithm rather improves on the former issue, limiting the over-estimation bias by using two critics and taking the lowest estimate of the action values in the update mechanisms.

## 4 METHODS

As shown in Figure 1a, the CEM-RL method combines CEM with either DDPG or TD3, giving rise to two algorithms named CEM-DDPG and CEM-TD3. The mean actor of the CEM population, referred to as $\pi_\mu$, is first initialized with a random actor network. A unique critic network $\mathcal{Q}^\pi$ managed by TD3 or DDPG is also initialized. At each iteration, a population of actors is sampled by adding Gaussian noise around the current mean $\pi_\mu$, according to the current covariance matrix $\Sigma$. Half of the resulting actors are directly evaluated. The corresponding fitness is computed as the cumulative reward obtained during an episode in the environment. Then, for each actor of the other half of the population, the critic is updated using this actor and, reciprocally, the actor follows the direction of the gradient given by the critic $\mathcal{Q}^\pi$ for a fixed number of steps. The resulting actors are evaluated after this process. The CEM algorithm then takes the top-performing half of the resulting global population to compute its new $\pi_\mu$ and $\Sigma$. The steps performed in the environment used to evaluate all actors in the population are fed into the replay buffer. The critic is trained from that buffer pro rata to the quantity of new information introduced in the buffer at the current generation. For instance, if the population contains 10 individuals, and if each episode lasts 1000 time steps, then 10,000 new samples are introduced in the replay buffer at each generation. The critic is thus trained

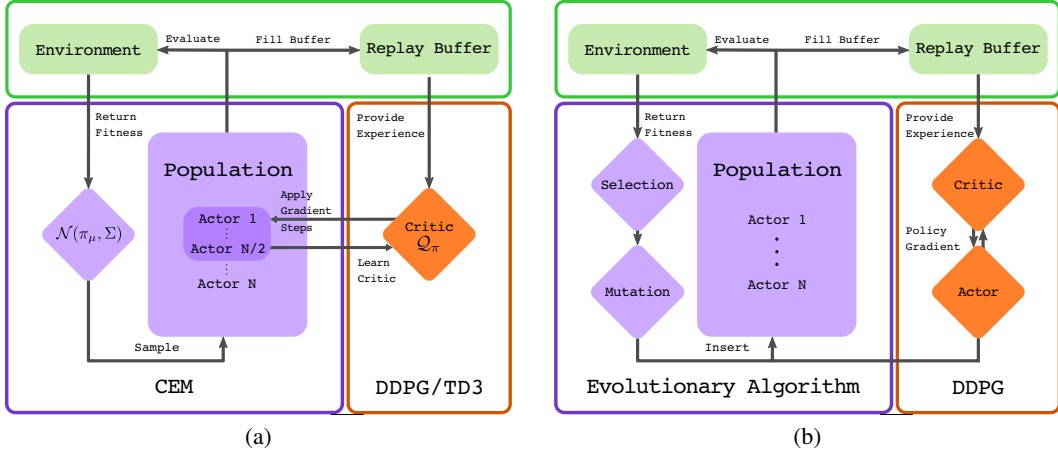

Figure 1: Architectures of the CEM-RL (a) and ERL (b) algorithms

for 10,000 mini-batches, which are divided into 2000 mini-batches per learning actor. This is a common practice in deep RL algorithms, where one mini-batch update is performed for each step of the actor in the environment. We also choose this number of steps (10,000) to be the number of gradient steps taken by half of the population at the next iteration. A pseudo-code of CEM-RL is provided in Algorithm 1.

In cases where applying the gradient increases the performance of the actor, CEM benefits from this increase by incorporating the corresponding actors in its computations. By contrast, in cases where the gradient steps decrease performance, the resulting actors are ignored by CEM, which instead focuses on standard samples around $\pi_\mu$. Those poor samples do not bring new insight on the current distribution of the CEM algorithm, since the gradient steps takes them away from the current distribution. However, since all evaluated actors are filling the replay buffer, the resulting experience is still fed to the critic and the future learning actors, providing some supplementary exploration.

This approach generates a beneficial flow of information between the deep RL part and the evolutionary part. Indeed, on one hand, good actors found by following the current critic directly improve the evolutionary population. On the other hand, good actors found through evolution fill the replay buffer from which the RL algorithm learns.

In that respect, our approach benefits from the same properties as the ERL algorithm (Khadka & Tumer, 2018b) depicted in Figure 1b. But, by contrast with Khadka & Tumer (2018b), gradient steps are directly applied to several samples, and using the CEM algorithm makes it possible to use importance mixing, as described in Appendix B. Another difference is that in CEM-RL gradient steps are applied at each iteration whereas in ERL, a deep RL actor is only injected to the population from time to time. One can also see from Figure 1 that, in contrast to ERL, CEM-RL does not use any deep RL actor. Other distinguishing properties between ERL and CEM-RL are discussed in the light of empirical results in Section 5.2.

Finally, given that CMA-ES is generally considered as more sophisticated than CEM, one may wonder why we did not use CMA-ES instead of CEM into the CEM-RL algorithm. Actually, the key contribution of CMA-ES with respect to CEM consists of the evolutionary path mechanism (see Section 3.2), but this mechanism results in some inertia in $\Sigma$ updates, which resists to the beneficial effect of applying RL gradient steps.

## 5 EXPERIMENTAL STUDY

In this section, we study the CEM-RL algorithm to answer the following questions:

- How does the performance of CEM-RL compare to that of CEM and TD3 taken separately? What if we remove the CEM mechanism, resulting in a multi-actor TD3?

---

**Algorithm 1** CEM-RL

---

**Require:** *max_steps*, the maximum number of steps in the environment
$\qquad\quad$ $\tau_{\text{CEM}}, \sigma_{init}, \sigma_{end}$ and *pop_size*, hyper-parameters of the CEM algorithm
$\qquad\quad$ $\gamma, \tau, lr_{actor}$ and $lr_{critic}$, hyper-parameters of DDPG

1: Initialize a random actor $\pi_\mu$, to be the mean of the CEM algorithm
2: Let $\Sigma = \sigma_{init}\mathcal{I}$ be the covariance matrix of the CEM algorithm
3: Initialize the critic $\mathcal{Q}^\pi$ and the target critic $\mathcal{Q}^\pi_t$
4: Initialize an empty cyclic replay buffer $\mathcal{R}$

5: *total_steps, actor_steps* $= 0, 0$
6: **while** *total_steps < max_steps*:

7: $\quad$ Draw the current population *pop* from $\mathcal{N}(\pi_\mu, \Sigma)$ with importance mixing (see Algorithm 2 in Appendix B)
8: $\quad$ **for** $i \leftarrow 1$ to *pop_size*/2:
9: $\qquad$ Set the current policy $\pi$ to *pop*[i]
10: $\qquad$ Initialize a target actor $\pi_t$ with the weights of $\pi$
11: $\qquad$ Train $\mathcal{Q}^\pi$ for 2 * *actor_steps* / *pop_size* mini-batches
12: $\qquad$ Train $\pi$ for *actor_steps* mini-batches
13: $\qquad$ Reintroduce the weights of $\pi$ in *pop*

14: $\quad$ *actor_steps* $= 0$
15: $\quad$ **for** $i \leftarrow 1$ to *pop_size*:
16: $\qquad$ Set the current policy $\pi$ to *pop*[i]
17: $\qquad$ (fitness $f$, steps $s$) $\leftarrow$ evaluate($\pi$)
18: $\qquad$ Fill $\mathcal{R}$ with the collected experiences
19: $\qquad$ *actor_steps* = *actor_steps* + $s$
$\qquad$ *total_steps* = *total_steps* + *actor_steps*

20: $\quad$ Update $\pi_\mu$ and $\Sigma$ with the top half of the population (see (1) and (2) in Section 3.2)

21: **end while**

---

- How does CEM-RL perform compared to ERL? What are the main factors explaining the difference between both algorithms?

Additionally, in Appendices B to E, we investigate other aspects of the performance of CEM-RL such as the impact of importance mixing, the addition of action noise or the use of the $tanh$ non-linearity.

## 5.1 EXPERIMENTAL SETUP

In order to investigate the above questions, we evaluate the corresponding algorithms in several continuous control tasks simulated with the MUJOCO physics engine and commonly used as policy search benchmarks: HALF-CHEETAH-V2, HOPPER-V2, WALKER2D-V2, SWIMMER-V2 and ANT-V2 (Brockman et al., 2016).

We implemented CEM-RL with the PYTORCH library [1]. We built our code around the DDPG and TD3 implementations given by the authors of the TD3 algorithm[2]. For the ERL implementation, we used one given by the authors[3].

Unless specified otherwise, each curve represents the average over 10 runs of the corresponding quantity, and the variance corresponds to the $68\%$ confidence interval for the estimation of the mean. In all learning performance figures, dotted curves represent medians and the x-axis represents the

---

[1]The code for reproducing the experiments is available at `https://github.com/apourchot/CEM-RL`.
[2]Available at `https://github.com/sfujim/TD3`.
[3]Available at `https://github.com/ShawK91/erl_paper_nips18`.

total number of steps actually performed in the environment, to highlight potential sample efficiency effects, particularly when using importance mixing (see Appendix B).

Architectures of the networks are described in Appendix A. Most TD3 and DDPG hyper-parameters were reused from Fujimoto et al. (2018). The only notable difference is the use of $tanh$ non linearities instead of RELU in the actor network, after we spotted that the latter performs better than the former on several environments. We trained the networks with the Adam optimizer (Kingma & Ba, 2014), with a learning rate of $1e^{-3}$ for both the actor and the critic. The discount rate $\gamma$ was set to 0.99, and the target weight $\tau$ to $5e^{-3}$. All populations contained 10 actors, and the standard deviations $\sigma_{init}$, $\sigma_{end}$ and the constant $\tau_{cem}$ of the CEM algorithm were respectively set to $1e^{-3}$, $1e^{-5}$ and 0.95. Finally, the size of the replay buffer was set to $1e^6$, and the batch size to 100.

## 5.2 RESULTS

We first compare CEM-TD3 to TD3, TD3 and a multi-actor variant of TD3, then CEM-RL to ERL based on several benchmarks. A third section is devoted to additional results which have been rejected in appendices to comply with space constraints.

### 5.2.1 COMPARISON TO CEM, TD3 AND A MULTI-ACTOR TD3

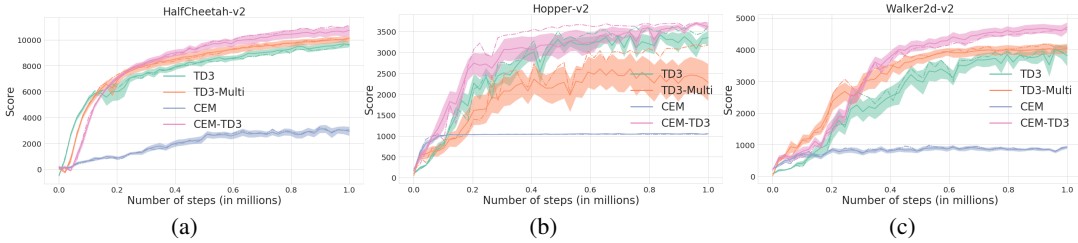

Figure 2: Learning curves of TD3, CEM and CEM-RL on the HALF-CHEETAH-V2, HOPPER-V2, and WALKER2D-V2 benchmarks.

In this section, we compare CEM-TD3 to three baselines: our variant of CEM, TD3 and a multi-actor variant of TD3. For TD3 and its multi-actor variant, we report the average of the score of the agent over 10 episodes for every 5000 steps performed in the environment. For CEM and CEM-TD3, we report after each generation the average of the score of the new average individual of the population over 10 episodes. From Figure 2, one can see that CEM-TD3 outperforms CEM and TD3 on HALF-CHEETAH-V2, HOPPER-V2 and WALKER2D-V2. On most benchmarks, CEM-TD3 also displays slightly less variance than TD3. Further results in Appendix G show that on ANT-V2, CEM-TD3 outperforms CEM and is on par with TD3. More surprisingly, CEM outperforms all other algorithms on SWIMMER-V2, as covered in Appendix E.

One may wonder whether the good performance of CEM-TD3 mostly comes from its "ensemble method" nature (Osband et al., 2016). Indeed, having a population of actors improves exploration and stabilizes performances by filtering out instabilities that can appear during learning. To answer this question, we performed an ablative study where we removed the CEM mechanism. We considered a population of 5 actors initialized as in CEM-TD3, but then just following the gradient given by the TD3 critic. This algorithm can be seen as a multi-actor TD3 where all actors share the same critic. We reused the hyper-parameters described in Section 5.2. From Figure 2, one can see that CEM-TD3 outperforms more or less significantly multi-actor TD3 on all benchmarks, which clearly suggests that the evolutionary part contributes to the performance of CEM-TD3.

As a summary, Table 1 gives the final performance of methods compared in this Section. We conclude that CEM-TD3 is generally superior to CEM, TD3 and multi-actor TD3. More precisely, in environments where TD3 provides a useful gradient information, CEM-TD3 enhances CEM by accelerating updates towards better actors, and it enhances TD3 by reducing variance in the learning process.

| Environment | CEM | | | TD3 | | |
|---|---|---|---|---|---|---|
| | Mean | Var. | Median | Mean | Var. | Median |
| HALF-CHEETAH-V2 | 2940 | 12% | 3045 | 9630 | 2.1% | 9606 |
| HOPPER-V2 | 1055 | 1.3% | 1040 | 3355 | 5.1% | 3626 |
| WALKER2D-V2 | 928 | 5.4% | 934 | 3808 | 8.9% | 3882 |
| SWIMMER-V2 | **351** | **2.7**% | **361** | 63 | 14% | 47 |
| ANT-V2 | 487 | 6.7% | 506 | 4027 | 10% | 4587 |

| Environment | TD3 Multi-Actor | | | CEM-TD3 | | |
|---|---|---|---|---|---|---|
| | Mean | Var. | Median | Mean | Var. | Median |
| HALF-CHEETAH-V2 | 9662 | 2.8% | 9710 | **10725** | **3.7**% | **11539** |
| HOPPER-V2 | 2056 | 20% | 2376 | **3613** | **2.9**% | **3722** |
| WALKER2D-V2 | 3934 | 4.1% | 3954 | **4711** | **3.3**% | **4637** |
| SWIMMER-V2 | 76 | 14% | 60 | 75 | 15% | 62 |
| ANT-V2 | 3567 | 22% | 3911 | **4251** | **5.9**% | **4310** |

Table 1: Final performance of CEM, TD3, multi-actor TD3 and CEM-TD3 on 5 environments. We report the mean ands medians over 10 runs of 1 million steps. For each benchmark, we highlight the results of the method with the best mean.

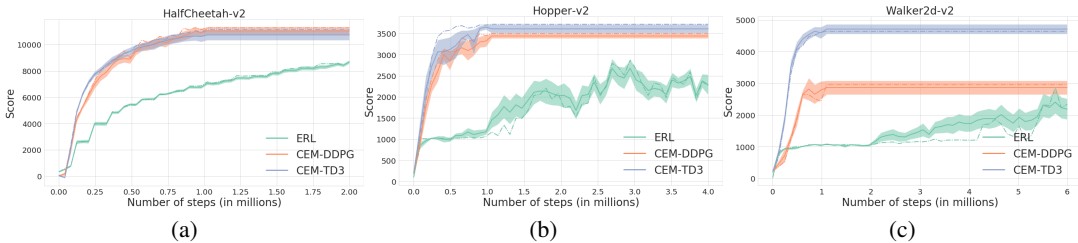

(a)             (b)             (c)

Figure 3: Learning curves of ERL, CEM-DDPG and CEM-TD3 on HALF-CHEETAH-V2, HOPPER-V2, ANT-V2 and WALKER2D-V2. Both CEM-RL methods are only trained 1 million steps.

### 5.2.2 COMPARISON TO ERL

In this section, we compare CEM-RL to ERL. The ERL method using DDPG rather than TD3, we compare it to both CEM-DDPG and CEM-TD3. This makes it possible to isolate the effect of the combination scheme from the improvement brought by TD3 itself. Results are shown in Figure 3. We let ERL learn for the same number of steps as in Khadka & Tumer, namely 2 millions on HALF-CHEETAH-V2 and SWIMMER-V2, 4 millions on HOPPER-V2, 6 millions on ANT-V2 and 10 millions on WALKER2D-V2. However, due to limited computational resources, we stop learning with both CEM-RL methods after 1 million steps, hence the constant performance after 1 million steps.

Our results slightly differ from those of the ERL paper (Khadka & Tumer, 2018b). We explain this difference by two factors. First, the authors only average their results over 5 different seeds, whereas we used 10 seeds. Second, the released implementation of ERL may be slightly different from the one used to produce the published results[4], raising again the reproducibility issue recently discussed in the reinforcement learning literature (Henderson et al., 2017).

Figure 3 shows that after performing 1 million steps, both CEM-RL methods outperform ERL on HALF-CHEETAH-V2, HOPPER-V2 and WALKER2D-V2. We can also see that CEM-TD3 outperforms CEM-DDPG on WALKER2D-V2. On ANT-V2, CEM-DDPG and ERL being on par after 1 million steps, we increased the number of learning steps in CEM-DDPG to 6 millions. The corresponding results are shown in Figure 10b in Appendix G. Results on SWIMMER-V2 are covered in Appendix E.

---

[4]personal communication with the authors

| Environment | ERL | | | CEM-DDPG | | | CEM-TD3 | | |
|---|---|---|---|---|---|---|---|---|---|
| | Mean | Var. | Median | Mean | Var. | Median | Mean | Var. | Median |
| HALF-CHEETAH-V2 | 8684 | 1.5% | 8675 | **11035** | **2.7%** | **11276** | 10725 | 3.7% | 11539 |
| HOPPER-V2 | 2288 | 10.5% | 2267 | 3444 | 1.6% | 3499 | **3613** | **2.9%** | **3722** |
| WALKER2D-V2 | 2188 | 15% | 2338 | 2865 | 7.6% | 2958 | **4711** | **3.3%** | **4637** |
| SWIMMER-V2 | **350** | **2.41%** | **360** | 268 | 12% | 279 | 75 | 15% | 62 |
| ANT-V2 | 3716 | 18.1% | 4240 | 2170 | 52% | 3574 | **4251** | **5.9%** | **4310** |

Table 2: Final performance of ERL, CEM-DDPG and CEM-TD3 on 5 environments. We report the mean ands medians over 10 runs of 1 million steps. For each benchmark, we highlight the results of the method with the best mean.

One can see that, beyond outperforming ERL, CEM-TD3 outperforms CEM-DDPG on most benchmarks, in terms of final performance, convergence speed, and learning stability. This is especially true for hard environments such as WALKER2D-V2 and ANT-V2. The only exception in SWIMMER-V2, as studied in Appendix E.

Table 2 gives the final best results of methods used in this Section. The overall conclusion is that CEM-RL generally outperforms ERL.

### 5.2.3 ADDITIONAL RESULTS

In this section, we outline the main messages arising from further studies that have been rejected in appendices in order to comply with space constraints.

In Appendix B, we investigate the influence of the importance mixing mechanism over the evolution of performance, for CEM and CEM-RL. Results show that importance mixing has a limited impact on the sample efficiency of CEM-TD3 on the benchmarks studied here, in contradiction with results from Pourchot et al. (2018) obtained using various standard evolutionary strategies. The fact that the covariance matrix $\Sigma$ moves faster with CEM-RL may explain this result, as it prevents the reuse of samples.

In Appendix C, we analyze the effect of adding Gaussian noise to the actions of CEM-TD3. Unlike what Khadka & Tumer (2018b) suggested using ERL, we did not find any conclusive evidence that action space noise improves performance with CEM-TD3. This may be due to the fact that, as further studied in Appendix D, the evolutionary algorithm in ERL tends to converge to a unique individual, hence additional noise is welcome, whereas evolutionary strategies like CEM more easily maintain some exploration. Indeed, we further investigate the different dynamics of parameter space exploration provided by the ERL and CEM-TD3 algorithms in Appendix D. Figure 6 and 7 show that the evolutionary population in ERL tends to collapse towards a single individual, which does not happen with the CEM population due to the sampling method.

In Appendix E, we highlight the fact that, on the SWIMMER-V2 benchmark, the performance of the algorithms studied in this paper varies a lot from the performance obtained on other benchmarks. The most likely explanation is that, in SWIMMER-V2, any deep RL method provides a deceptive gradient information which is detrimental to convergence towards efficient actor parameters. In this particular context, ERL better resists to detrimental gradients than CEM-RL, which suggests to design a version of ERL using CEM to improve the population instead of its ad hoc evolutionary algorithm.

Finally, in Appendix F, we show that using a $tanh$ non-linearity in the architecture of actors often results in significantly stronger performance than using RELU. This strongly suggests performing "neural architecture search" (Zoph & Le, 2016; Elsken et al., 2018) in the context of RL.

## 6 CONCLUSION AND FUTURE WORK

We advocated in this paper for combining evolutionary and deep RL methods rather than opposing them. In particular, we have proposed such a combination, the CEM-RL method, and showed that in most cases it was outperforming not only some evolution strategies and some sample efficient off-policy deep RL algorithms, but also another combination, the ERL algorithm. Importantly, despite

being mainly an evolutionary method, CEM-RL is competitive to the state-of-the-art even when considering sample efficiency, which is not the case of other deep neuroevolution methods (Salimans et al., 2017; Petroski Such et al., 2017).

Beyond these positive performance results, our study raises more fundamental questions. First, why does the simple CEM algorithm perform so well on the SWIMMER-V2 benchmark? Then, our empirical study of importance mixing did not confirm a clear benefit of using it, neither did the effect of adding noise on actions. We suggest explanations for these phenomena, but nailing down the fundamental reasons behind them will require further investigations. Such deeper studies will also help understand which properties are critical in the performance and sample efficiency of policy search algorithms, and define even more efficient policy search algorithms in the future. As suggested in Section 5.2.3, another avenue for future work will consist in designing an ERL algorithm based on CEM rather than on an ad hoc evolutionary algorithm. Finally, given the impact of the neural architecture on our results, we believe that a more systemic search of architectures through techniques such as neural architecture search (Zoph & Le, 2016; Elsken et al., 2018) may provide important progress in performance of deep policy search algorithms.

## 7 ACKNOWLEDGMENTS

This work was supported by the European Commission, within the DREAM project, and has received funding from the European Union's Horizon 2020 research and innovation program under grant agreement $N^o$ 640891. We would like to thank Thomas Pierrot for fruitful discussions.

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

## A   ARCHITECTURE OF THE NETWORKS

Our network architectures are very similar to the ones described in Fujimoto et al. (2018). In particular, the size of the layers remains the same. The only differences resides in the non-linearities. We use tanh operations in the actor between each layer, where Fujimoto et al. use RELU and we use leaky RELU in the critic, where Fujimoto et al. use simple RELU. Reasons for this choice are presented in Appendix F.

Table 3: Architecture of the networks (from the input layer (top line) to the output layer (bottom line)

| Actor | Critic |
|-------|--------|
| (state dim, 400) | (state dim + action dim, 400) |
| tanh | leaky RELU |
| (400, 300) | (400, 300) |
| tanh | leaky RELU |
| (300, action dim) | (300, 1) |
| tanh | |

## B   IMPORTANCE MIXING

Importance mixing is a specific mechanism designed to improve the sample efficiency of evolution strategies. It was initially introduced in Sun et al. (2009) and consisted in reusing some samples from the previous generation into the current one, to avoid the cost of re-evaluating the corresponding policies in the environment. The mechanism was recently extended in Pourchot et al. (2018) to reusing samples from any generation stored into an archive. Empirical results showed that importance sampling can improve sample efficiency by a factor of ten, and that most of these savings just come from using the samples from the previous generation, as performed by the initial mechanism. A pseudo-code of the importance mixing mechanism is given in Algorithm 2.

In CEM, importance mixing is implemented as described in (Pourchot et al., 2018). By contrast, some adaptation is required in CEM-RL. Actors which take gradient steps can no longer be regarded as sampled from the current distribution of the CEM algorithm. We thus choose to apply importance mixing only to the half of the population which does not receive gradient steps from the RL critic. In practice, only actors which do not take gradient steps are inserted into the actor archive and can be replaced with samples from previous generations.

From Figure 4, one can see that in the CEM case, importance mixing introduces some minor instability, without noticeably increasing sample efficiency. On HALF-CHEETAH-V2, SWIMMER-V2 and WALKER2D-V2, performance even decreases when using importance mixing. For CEM-RL, the effect varies greatly from environment to environment, but the gain in sample reuse is almost null

---

**Algorithm 2** Importance mixing

---

**Require:** $p(z, \boldsymbol{\theta}_{new})$: current probability density function (pdf), $p(z, \boldsymbol{\theta}_{old})$: old pdf, $g_{old}$: old generation

1: $g_{new} \leftarrow \emptyset$
2: **for** $i \leftarrow 1$ to $N$

3:     Draw rand1 and rand2 uniformly from $[0, 1]$

4:     Let $z_i$ be the $i^{th}$ individual of the old generation $g_{old}$
5:     **if** $min(1, \frac{p(z_i, \boldsymbol{\theta}_{new})}{p(z_i, \boldsymbol{\theta}_{old})}) >$ rand1:
6:         Append $z_i$ to the current generation $g_{new}$

7:     Draw $z'_i$ according to the current pdf $p(., \boldsymbol{\theta}_{new})$
8:     **if** $max(0, 1 - \frac{p(z'_i, \boldsymbol{\theta}_{old})}{p(z'_i, \boldsymbol{\theta}_{new})}) >$ rand2:
9:         Append $z'_i$ to the current generation $g_{new}$

10:     size $= |g_{new}|$
11:     **if** size $\geq N$: go to 12

12: **if** size $> N$: remove a randomly chosen sample
13: **if** size $< N$: fill the generation sampling from $p(., \boldsymbol{\theta}_{new})$
14: **return** $g_{new}$

---

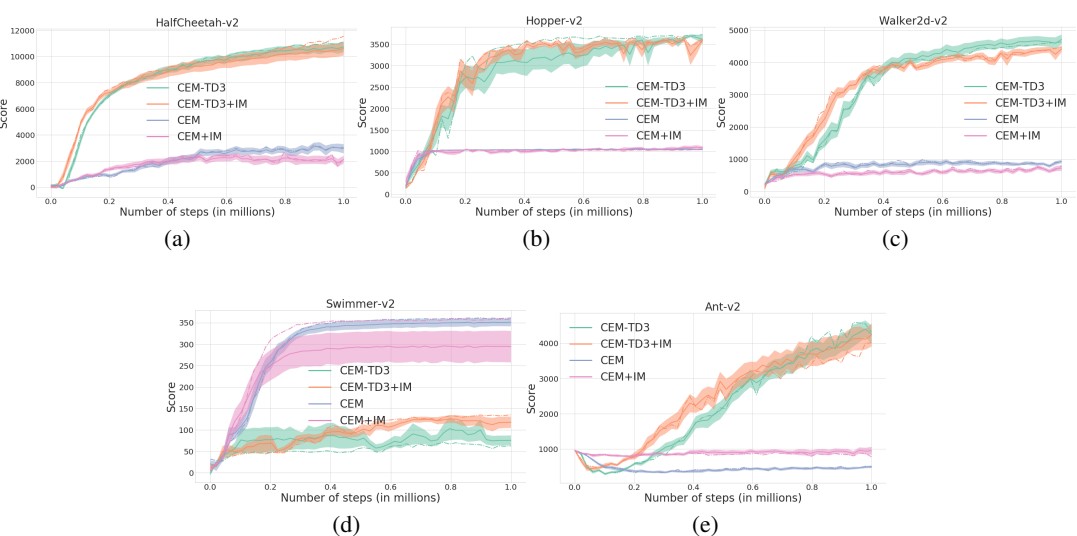

Figure 4: Learning curves of CEM-TD3 and CEM with and without importance mixing on the HALF-CHEETAH-V2, HOPPER-V2, WALKER2D-V2, SWIMMER-V2 and ANT-V2 benchmarks.

as well, though an increase in performance can be seen on SWIMMER-V2. The latter fact is consistent with the finding that the gradient steps are not useful in this environment (see Appendix E). On HOPPER-V2 and HALF-CHEETAH-V2, results with and without importance mixing seem to be equivalent. On WALKER2D-V2, importance mixing decreases final performance. On ANT-V2, importance mixing seems to accelerate learning in the beginning, but final performances are equivalent to those of CEM-RL. Thus importance mixing seems to have a limited impact in CEM-TD3.

This conclusion seems to contradict the results obtained in Pourchot et al. (2018). This may be due to different things. First, the dimensions of the search spaces in the experiments here are much larger than those studied in Pourchot et al. (2018), which might deteriorate the estimation of the covariance matrices when samples are too correlated. On top of this, the MUJOCO environments are harder than

| Environment | CEM-TD3 | | | CEM-TD3 + IM | | |
|---|---|---|---|---|---|---|
| | Mean | Var. | Median | Mean | Var. | Median |
| HALF-CHEETAH-V2 | **10725** | **3.7**% | **11147** | 10601 | 4.9% | 11539 |
| HOPPER-V2 | **3613** | **2.9**% | **3722** | 3589 | 1.2% | 3616 |
| WALKER2D-V2 | **4711** | **3.3**% | **4637** | 4420 | 2.3% | 4468 |
| SWIMMER-V2 | 75 | 15% | 62 | **117** | **11**% | **135** |
| ANT-V2 | **4251** | **5.9**% | **4310** | 4235 | 7.8% | 4013 |

Table 4: Final performance of CEM-TD3 with and without importance mixing on the HALF-CHEETAH-V2, HOPPER-V2, SWIMMER-V2, ANT-V2 and WALKER2D-V2 environments. We report the mean ands medians over 10 runs of 1 million steps. For each benchmark, we highlight the results of the method with the best mean.

the ones used in Pourchot et al. (2018). In particular, we can see from Figure 2 that CEM is far from solving the environments over one million steps. Perhaps a study over a longer time period would make importance mixing relevant again. Besides, by reusing old samples, the importance mixing mechanism somehow hinders exploration (since we evaluate less new individuals), which might be detrimental in the case of MUJOCO environments. Finally, and most importantly, the use of RL gradient steps accelerates the displacement of the covariance matrix, resulting in fewer opportunities for sample reuse.

## C    EFFECT OF ACTION NOISE

In Khadka & Tumer (2018b), the authors indicate that one reason for the efficiency of their approach is that the replay buffer of DDPG gets filled with two types of noisy experiences. On one hand, the buffer gets filled with noisy interactions of the DDPG actor with the environment. This is usually referred to as *action space noise*. On the other hand, actors with different parameters also fill the buffer, which is more similar to *parameter space noise* (Plappert et al., 2017). In CEM-RL, we only use parameter space noise, but it would also be possible to add action space noise. To explore this direction, each actor taking gradient steps performs a noisy episode in the environment. We report final results after 1 million steps in Table 5. Learning curves are available in Figure 5.

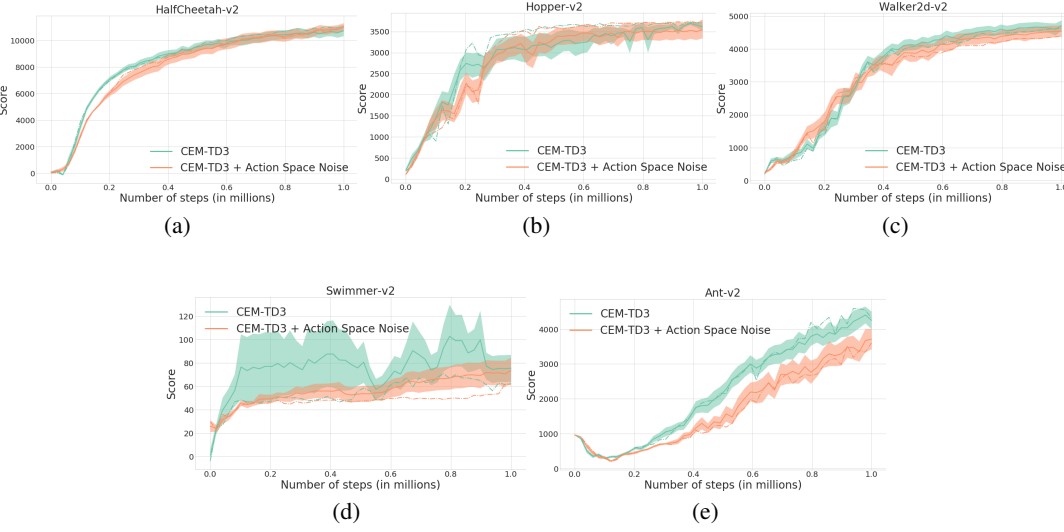

Figure 5: Learning curves of CEM-RL with and without action space noise on the HALF-CHEETAH-V2, HOPPER-V2, WALKER2D-V2, SWIMMER-V2 and ANT-V2 benchmarks.

Unlike what Khadka & Tumer (2018b) suggested, we did not find any conclusive evidence that action space noise improves performance. In CEM-TD3, the CEM part seems to explore enough of the action space on its own. It seems that sampling performed in CEM results in sufficient exploration and performs better than adding simple Gaussian noise to the actions. This highlights a difference between using an evolutionary strategy like CEM and an evolutionary algorithm as done in ERL. Evolutionary algorithms tend to converge to a unique individual whereas evolutionary strategies more easily maintain some exploration. These aspects are further studied in Appendix D.

| | CEM-TD3 | | | CEM-TD3 + AN | | | CEM-TD3- RELU | | |
|---|---|---|---|---|---|---|---|---|---|
| Environment | Mean | Var. | Median | Mean | Var. | Median | Mean | Var. | Median |
| HALF-CHEETAH-V2 | **10725** | **3.7**% | **11147** | 11006 | 2.7% | 11086 | 10267 | 3.7% | 10133 |
| HOPPER-V2 | **3613** | **2.9**% | **3722** | 3541 | 5.7% | 3761 | 3604 | 2.6% | 3716 |
| WALKER2D-V2 | **4711** | **3.3**% | **4637** | 4542 | 5.7% | 4392 | 4311 | 7.5% | 4534 |
| SWIMMER-V2 | 75 | 15% | 62 | 74 | 15% | 62 | **118** | **21**% | **114** |
| ANT-V2 | **4251** | **5.9**% | **4310** | 3711 | 7.9% | 3604 | 2264 | 16% | 2499 |

Table 5: Final Performance of CEM-RL with and without action noise (AN), with DDPG, and with RELU non-linearities in MUJOCO environments. We report the mean ands medians over 10 runs of 1 million steps. For each benchmark, we highlight the results of the method with the best mean.

## D  PARAMETER SPACE EXPLORATION IN CEM-RL AND ERL

In this section, we highlight the difference in policy parameter update dynamics in CEM-RL and ERL. Figure 6 displays the evolution of the first two parameters of actor networks during training with CEM-RL and ERL on HALF-CHEETAH-V2. For ERL, we plot the chosen parameters of the DDPG actor with a continuous line, and represent those of the evolutionary actors with dots. For CEM-RL, we represent the chosen parameters of sampled actors with dots, and the gradient steps based on the TD3 critic with continuous lines. The same number of dots is represented for both algorithms.

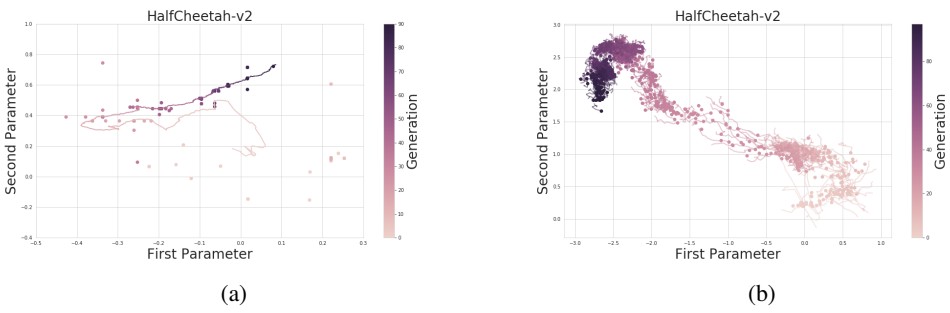

(a)                                                                 (b)

Figure 6: Evolution of the first two parameters of the actors when learning with (a) ERL and (b) CEM-TD3. Dots are sampled parameters of the population and continuous lines represent parameters moved through RL gradient steps.

One can see that, in ERL the evolutionary population tends to be much less diverse that in CEM-RL. There are many redundancies in the parameters (dots with the same coordinates), and the population seems to converge to a single individual. On the other hand, there is no such behavior in CEM-RL where each generation introduces completely new samples. As a consequence, parameter space exploration looks better in the CEM-RL algorithm.

To further study this loss of intra-population diversity in ERL, we perform 10 ERL runs and report in Figure 7 an histogram displaying the distribution of the population-wise similarity with respect to the populations encountered during learning. We measure this similarity as the average percentage of parameters shared between two different individuals of the said population. The results indicate that around 55% of populations encountered during a run of ERL display a population-similarity of above 80%.

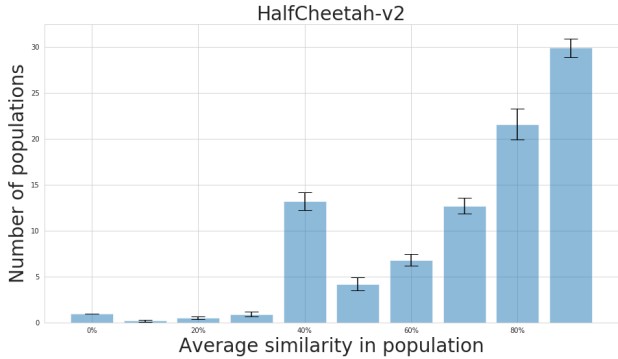

Figure 7: Histogram of the average similarity in populations during learning with the ERL algorithm. Results are averaged over 10 runs. As usual, the variance corresponds to the 68% confidence interval for the estimation of the mean.

One can also see the difference in how both methods use the gradient information of their respective deep RL part. In the case of ERL, the parameters of the population concentrate around those of the DDPG actor. Each 10 generations, its parameters are introduced into the population, and since DDPG is already efficient alone on HALF-CHEETAH-V2, those parameters quickly spread into the population. Indeed, according to Khadka & Tumer (2018b), the resulting DDPG actor is the elite of the population 80% of the time, and is introduced into the population 98% of the time. This integration is however passive: the direction of exploration does not vary much after introducing the DDPG agent. CEM-RL integrates this gradient information differently. The short lines emerging from dots, which represent gradient steps performed by half of the actors, act as scouts. Once CEM becomes aware of better solutions that can be found in a given direction, the sampling of the next population is modified so as to favor this promising direction. CEM is thus pro-actively exploring in the good directions it has been fed with.

## E  THE CASE OF THE SWIMMER-V2 BENCHMARK

Experiments on the SWIMMER-V2 benchmark give results that differ a lot from the results on other benchmarks, hence they are covered separately here. Figure8a shows that CEM outperforms TD3, CEM-TD3, multi-actor TD3. Besides, as shown in Figure 8b, ERL outperforms CEM-DDPG, which itself outperforms CEM-TD3.

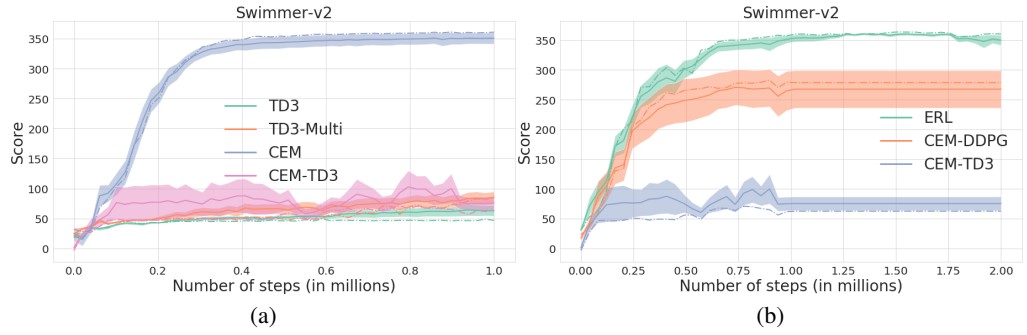

Figure 8: Learning curves on the SWIMMER-V2 environment of (a): CEM and TD3, multi-actor TD3 and CEM-RL; (b) ERL, CEM-DDPG and CEM-TD3.

All these findings seem to show that being better at RL makes you worse at SWIMMER-V2. The most likely explanation is that, in SWIMMER-V2, any deep RL method provides a deceptive gradient information which is detrimental to convergence towards efficient actor parameters. This conclusion

could already be established from the results of Khadka & Tumer (2018b), where the evolution algorithm alone produced results on par with the ERL algorithm, showing that RL-based actors were just ignored. In this particular context, the actors using TD3 gradient being deteriorated by the deceptive gradient effect, CEM-RL is behaving as a CEM with only half a population, thus it is less efficient than the standard CEM algorithm. By contrast, ERL better resists than CEM-RL to the same issue. Indeed, if the actor generated by DDPG does not perform better than the evolutionary population, then this actor is just ignored, and the evolutionary part behaves as usual, without any loss in performance. In practice, Khadka & Tumer note that on SWIMMER-V2, the DDPG actor was rejected 76% of the time. Finally, by comparing CEM and ERL from Figure 8a and Figure 8b, one can conclude that on this benchmark, the evolutionary part of ERL behaves on par with CEM alone. This is at odds with premature convergence effects seen in the evolutionary part of ERL, as studied in more details in Appendix D. From all these insights, the SWIMMER-V2 environment appears particularly interesting, as we are not aware of any deep RL method capable of solving it quickly and reliably.

## F  USING THE RELU OR TANH NON-LINEARITY

In this section, we explore the impact on performance of the type of non-linearities used in the actor of CEM-TD3. Table 5 reports the results of CEM-TD3 using RELU non-linearities between the linear layers, instead of $tanh$.

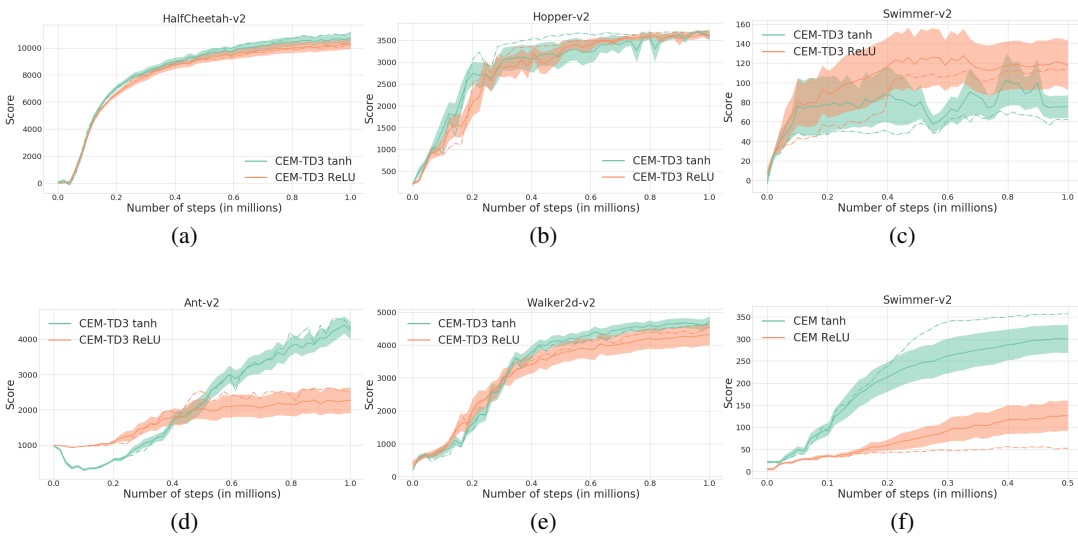

Figure 9: Learning curves of CEM-RL with $tanh$ and RELU as non-linearities in the actors, on the (a) HALF-CHEETAH-V2, (b) HOPPER-V2, (c) SWIMMER-V2, (d) ANT-V2 and (e) WALKER2D-V2 benchmarks. (f) shows the same of CEM on the SWIMMER-V2 benchmark.

Figure 9 displays the learning performance of CEM-TD3 and CEM on benchmarks, using either the RELU or the $tanh$ nonlinearity in the actors. Results indicate that on some benchmarks, changing from $tanh$ to RELU can cause a huge drop in performance. This is particularly obvious in the ANT-V2 benchmark, where the average performance drops by 46%. Figure 9(f) shows that, for the CEM algorithm on the SWIMMER-V2 benchmark, using RELU also causes a 60% performance drop. As previously reported in the literature (Henderson et al., 2017), this study suggests that network architectures can have a large impact on performance.

## G  ADDITIONAL RESULTS ON ANT-V2

Figure 10 represents the learning performance of CEM, TD3, multi-actor TD3, CEM-DDPG and CEM-TD3 on the ANT-V2 benchmark. It is discussed in the main text.

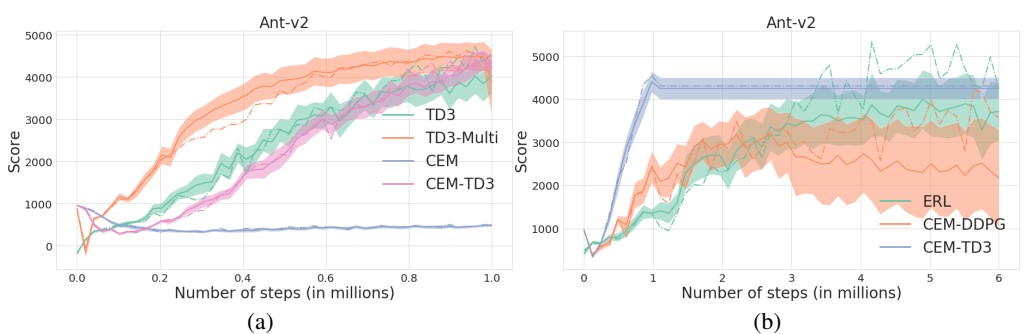

(a)  (b)

Figure 10: Learning curves of CEM-RL, CEM and TD3 on the SWIMMER-V2 and ANT-V2 benchmarks.

