# OpenReview forum: "CEM-RL: Combining evolutionary and gradient-based methods for policy search"
_ICLR.cc/2019/Conference_

### Official Review · AnonReviewer2 · 2018-11-02
**In total, the paper offers limited novelty. The results are good but the added margin to TD3 is rather small. I would therefore see the paper at the borderline.**

**Rating:** 7
**Confidence:** 5

**Review:**

The paper presents a combination of evolutionary search methods (CEM) and deep reinforcement learning methods (TD3). The CEM algorithm is used to learn a Diagional Gaussian distribution over the parametes of the policy. The population is sampled from the distribution. Half of the population is updated by the TD3 gradient before evaluating the samples. For filling the replay buffer of TD3, all state action samples from all members of the population are used. The algorithm is compared against the plane variants of CEM and TD3 as well as against the evoluationary RL (ERL) algorithm. Results are promising with a negative result on the swimmer_v2 task.

The paper is well written and easy to understand. While the presented ideas are well motivated and it is certainly a good idea to combine deep RL and evoluationary search, novelty of the approach is limited as the setup is quite similar to the ERL algorithm (which is still on archive and not published, but still...). See below for more comments:
- While there seems to be a consistent improvement over TD3, this improvement is in some cases small (e,g. ants).
- We are learning a value function for each of the first half of the population. However, the value function from the previous individual is used to initialize the learning of the current value function. Does this cause some issues, e.g., do we need to set the number of steps so high that the initialization does not matter so much any more? Or would it make more sense to reset the value function to some "mean value function" after every individual?
- The importance mixing does not seem to provide a better performance and could therefore be shortened in the paper

---

> ### Author Response · Authors · 2018-11-20
> **Response to Reviewer 2**
>
> We thank the reviewer for raising useful points which helped us a lot improving the paper.
>
> The main point of the reviewer is that the novelty of our approach is limited with respect to the Evolutionary RL (ERL) algorithm, and that improvement is sometimes small. These remarks helped us realize that we had to better highlight the differences between our approach and ERL, both in terms of concepts and performance. We did so by replacing Figure 1, which was contrasting CEM-RL to CEM, with a figure directly contrasting CEM-RL to ERL. We also added Figure 6 which better highlights the properties of the algorithms and we performed several additional studies, described either in the main text or in appendices.
>
> By the way, the ERL paper is now published at NIPS, but it was not the case yet when we submitted ours. We updated the corresponding reference.
>
> The reviewer seems to consider that each actor in our CEM-RL algorithm comes with its own critic (the reviewer says value function), which would raise a value function initialization issue. Actually, this is not the case: there is a single TD3 critic over the whole process, and gradient steps are applied to all the selected actors from that single critic. This has been clarified in the text by insisting on the unicity of this critic.
>
> We agree with the reviewer that the importance mixing did not provide the sample efficiency improvement we expected, and we can only provide putative explanations of why so far. Nevertheless, we believe this mechanism still has some potential and is currently overlooked by most deep neuroevolution researchers, so we decided to keep the importance mixing study in Appendix B rather than just removing it.

---

### Official Review · AnonReviewer3 · 2018-11-02
**Interesting result, missing a control**

**Rating:** 7
**Confidence:** 4

**Review:**

Gradient-free evolutionary search methods for Reinforcement Learning are typically very stable, but scale poorly with the number of parameters when optimizing highly-parametrized policies (e.g. neural networks). Meanwhile, gradient-based deep RL methods, such as DDPG are often sample efficient, particularly in the off-policy setting when, unlike evolutionary search methods, they can continue to use previous experience to estimate values. However, these approaches can also be unstable.

This work combines the well-known CEM search with TD3 (an improved variant of DDPG). The key idea of of this work is in each generation of CEM, 1/2 the individuals are improved using TD3 (i.e. the RL gradient). This method is made more practical by using a replay buffer so experience from previous generations is used for the TD3 updates and importance sampling is used to improve the efficiency of CEM.

This work shows, on some simple control tasks, that this method appears to result in much stronger performance compared with CEM, and small improvements over TD3 alone. It also typically out-performs ERL.

Intuitively, it seems like it may be possible to construct counter-examples where the gradient updates will prevent convergence. Issues of convergence seem like they deserve some discussion here and potentially could be examined empirically (is CEM-TD3 converging in the swimmer?).

The justification that the method of Khadka & Tumer (2018) cannot be extended to use CEM, since the RL policies do not comply with the covariance matrix is unclear to me. Algorithm 1, step 20, the covariance matrix is updated after the RL step so regardless of how the RL policies are generated, the search distribution on the next distribution includes them. In both this work, and Khadka & Tumer, the RL updates lead to policies that differ from the search distribution (indeed that is the point), and there is no guarantee in this work that the TD3 updates result in policies close to the starting point. It sees like the more important distinction is that, in this approach, the information flows both from ES to RL and vice-versa, rather than just from RL to ES.

One view of this method would be that it is an ensemble method for learning the policy [e.g. similar to Osband et al., 2016 for DQN]. This could be discussed and a relevant control would be to keep a population (ensemble) of policies, but only update using RL while sharing experience across all actors. This would isolate the ensemble effect from the evolutionary search.

Minor issues:

- The ReLU non-linearity in DDPG and TD3 prior work is replaced with tanh. This change is noted, but it would be useful to make a least a brief (i.e. one sentence) comment on the motivation for this change.

- The paper is over the hard page limit for ICLR so needs to be edit to reduce the length.

Osband I, Blundell C, Pritzel A, Van Roy B. Deep exploration via bootstrapped DQN. InAdvances in neural information processing systems 2016 (pp. 4026-4034).

---

> ### Author Response · Authors · 2018-11-20
> **Response to Reviewer 3 (1/2)**
>
> We thank the reviewer for his/her positive evaluation of our paper and for raising many very useful points which helped us getting to a clearer picture of our contribution. A few of these points deserve discussion beyond the changes made in the paper.
>
> Due to a mistake on page 2, we got the reviewer confused believing we are using importance sampling while we are using importance mixing instead. This has been fixed.
>
> The reviewer mentions it may be possible to construct counter-examples where the gradient updates will prevent convergence. This is a very important point. There are many RL problems (see e.g. Continuous Mountain Car, Colas et al. at ICML 2018) where at some point the gradient computed by the critic is deceptive, i.e. it drives the policy parameters into a wrong direction. In that case, applying that gradient to CEM actors as we do in CEM-RL is counter-productive. But the fact that we only apply this gradient to half the population makes it that CEM-RL should nevertheless overcome this issue:  the actors which did not receive a gradient step will be selected and the population will continue improving. However, admittedly, in this very specific context, CEM-RL is behaving as a CEM with only half a population, thus it is less efficient than the standard CEM. Besides, ERL even better resists than our approach to the same issue: if the actor generated by DDPG does not perform better than the evolutionary population due to a deceptive gradient issue, then this actor is just ignored, and the evolutionary part behaves as usual, without any loss in performance. This deceptive gradient issue certainly explains why CEM is the best approach on Swimmer. Finally, it may also happen that the RL part does not bring benefit just because the current critic is wrong and provides an inadequate gradient, in a non-deceptive gradient case. All the above points have now been made much clear in the new version of the paper, in particular we added an appendix dedicated to the swimmer benchmark.
>
> The reviewer also raises doubts about the fact that the method of Khadka & Tumer (2018) cannot be extended to use CEM. After second thoughts, this is absolutely right. As the reviewer says, in both this work and Khadka & Tumer, the RL updates lead to policies that may differ a lot from the search distribution and there is no guarantee in this work that the TD3 updates result in policies close to the starting point.
> But if the RL actor shows good enough performance, this does not prevent from computing a new covariance matrix which includes it. The corresponding ellipsoid in the search space may be very large, leading to a widespread next generation, but the process should tend to converge again towards a population of actors where evolutionary and RL actors are closer to each other.
>
> A result of these second thoughts is that one could definitely build an ERL algorithm where the evolutionary part is replaced by CEM. We corrected the paper according to this new insight. Unfortunately we did not find enough time to implement and test this algorithm during the rebuttal stage, but we now mention this possibility as an interesting avenue for future work.
>
> Despite the very interesting points above, the reviewer is wrong when saying that the main distinction between our approach and the ERL approach is that only in ours the information flow is from ES to RL and vice-versa. Actually, in ERL, if the RL actor added to the population performs well, it will steer the whole evolutionary population to the right direction just by generating offsprings, so RL and ES also benefit from each other.

---

> > ### Author Response · Authors · 2018-11-20
> > **Response to Reviewer 3 (2/2)**
> >
> > A lot of our effort during the rebuttal stage has been focused on better highlighting the often subtle differences between ERL and our approach. For doing so, we replaced Figure 1 with a figure directly contrasting CEM-RL to ERL. We also added Figure 6 which better highlights the properties of the algorithms and we performed several additional studies described either in the main text or in appendices.
> >
> > The next point of the reviewer is that a good deal of the strong performance of our method and RL may just be due to the fact that we are using multiple actors, thus benefiting from an "ensemble method" effect already mentioned in several papers such as Osband et al., 2016 for DQN. This point is absolutely valid.
> >
> > The reviewer thus suggests a relevant control which would be to keep a population (ensemble) of policies, but only update using RL while sharing experience across all actors. This would isolate the ensemble effect from the evolutionary search effect. We performed the suggested control. The resulting algorithm is a multiple-actor version of TD3. Results show that CEM-TD3 actually outperforms this multiple-actor TD3, thus the CEM part actually brings performance improvement.
> >
> > About replacing the ReLU non-linearity in DDPG and TD3 prior work with tanh, we spotted that we could get much better results on several environments with the latter. This explanation is now clearly mentioned in the paper, and motivates a future work direction which consists in using "neural architecture search" for RL problems, the performance of algorithms being a lot dependent on such architecture details.
> >
> > Finally, to keep our paper shorter than the hard page limit for ICLR while addressing all the reviewers points, we had to move several studies into appendices, starting with the importance mixing study.

---

> > > ### Comment · AnonReviewer3 · 2018-11-26
> > > **Addressed issues I raise**
> > >
> > > The revised version addresses the issues I raised. Thank you.
> > >
> > > I was already rating the paper positively, so my rating is unchanged.

---

### Official Review · AnonReviewer1 · 2018-11-02
**This paper proposes a combination scheme using cross-entropy method (CEM) and Twin Delayed Deep Deterministic Policy Gradients  (TD3), a policy of deep RL algorithm which improves over Deterministic Policy Gradients (DDPG).**

**Rating:** 6
**Confidence:** 3

**Review:**

The contributions of this paper are in the domain of policy search, where the authors combine evolutionary and gradient-based methods. Particularly, they propose a combination approach based on cross-entropy method (CEM) and TD3 as an alternative to existing combinations using either a standard evolutionary algorithm or a goal exploration process in tandem with the DDPG algorithm. Then, they show that CEM-RL has several advantages compared to its competitors and provides a satisfactory trade-off between performance and sample efficiency.

The authors evaluate the resulting algorithm, CEM-RL, using a set of benchmarks well established in deep RL, and they show that CEM-RL benefits from several advantages over its competitors and offers a satisfactory trade-off between performance and sample efficiency.  It is a pity to see that the authors provide acronyms without explicitly explaining them such as DDPG and TD3, and this right from the abstract.

The parer is  in general interesting, however the clarity of the paper is hindered  by the existence of several typos, and the writing in certain passages can be improved. Example of typos include  “an surrogate gradient”, “"an hybrid algorithm”,  “most fit individuals are used ” and so on…

In the related work the authors present the connection between their work and contribution to the state of the art in a detailed manner.  Similarly, in section 3 the authors provide an extensive background allowing to understand their proposed method.

In equation 1, 2 the updates of  \mu_new and \sigma_new uses \lambda_i, however the authors provide common choices for \lambda without any justification or references.

The proposed method is clearly explained and seems convincing. However the theoretical contribution is poor. And the experiment uses a very classical benchmark providing simulated data.

1. In the experimental study, the authors present the value of their tuning parameters (learning rate, target rate, discount rate…) at the initialisation phase without any justifications. And the experiments are limited to simulated data obtained from MUJOCO physics engine - a very classical benchmark.
2. Although the experiments are detailed and interesting they support poor theoretical developments and use a very classical benchmark

---

> ### Author Response · Authors · 2018-11-20
> **Response to Reviewer 1**
>
> We thank the reviewer for many positive comments about our paper.
>
> The typos explicitly mentioned in the review have been corrected, and we did our best to spot other typos not mentioned. Besides, all the acronyms have been explained.
>
> We added the tutorial from Hansen (2016) as the reference for the common choices for setting \lambda_i in Equation 1, 2.
>
> We agree with the reviewer that our paper is not theoretically oriented, nor does it address any real world application like robotics or other challenging domain. Our point is rather to provide a practical method performing well with respect to the state of the art, which is most often evaluated with the same widely used benchmarks.
>
> With respect to initialization of hyperparameters, as explicitly mentioned in the "experimental setup" section, "Most of the TD3 and DDPG hyper-parameters were reused from Fujimoto et al. (2018)." The justification for this choice is to facilitate comparison with previously published work.

---

> ### Comment · AnonReviewer1 · 2018-11-27
> **GIven all the efforts made by the authrors to substantially improve the quality of their papers I am reconsidering my rating.**
>
> The rebuttal provided by the authors is convincing.

---

### Author Response · Authors · 2018-11-20
**General overview of changes made**

We warmly thank the three reviewers for their valuable feedback on our paper. Their reviews helped us realize that the main weakness of our paper was insufficiently clear outline of the differences between our contribution and the Evolutionary RL (ERL) algorithm. As a consequence, we made many changes. We would appreciate if the reviewers could take a look at the new version of the paper before eventually revising their score or expressing further concerns.

The main changes we made with respect to the submitted version are the following:

* we added the control experiment suggested by Reviewer 3, to compare our CEM-RL framework with a 10 actors TD3 framework.

* we better outlined the conceptual differences between ERL and CEM-RL with several additional figures and a more thorough discussion: we replaced Figure 1 with a figure directly contrasting CEM-RL to ERL, we added Figure 6 which better highlights the properties of the algorithms and we performed several additional studies described either in the main text or in appendices.

* we have consolidated all experimental results, with longer runs when necessary.

* we have reorganized the paper, moving several side studies to appendices. In particular, following the suggestion of Reviewer 2, we moved the presentation and experimental study of importance mixing to Appendix B, leaving more room for comparison to ERL.

---

### Meta-Review · Area_Chair1 · 2018-12-13
**Simple method. Good results. Limited Novelty.**

**Confidence:** 4
**Recommendation:** Accept (Poster)

**Metareview:**

This paper combines two different types of existing optimization methods, CEM/CMA-ES and DDPG/TD3, for policy optimization. The approach resembles ERL but demonstrates good better performance on a variety of continuous control benchmarks.  Although I feel the novelty of the paper is limited, the provided promising results may justify the acceptance of the paper.